

# Foliar microbiome transplants confer disease resistance in a critically-endangered plant

Geoffrey Zahn[1] and Anthony S. Amend[2]

[1] Biology Department, Utah Valley University, Orem, UT, United States of America
[2] Botany Department, University of Hawaii at Manoa, Honolulu, HI, United States of America

## ABSTRACT

There has been very little effort to incorporate foliar microbiomes into plant conservation efforts even though foliar endophytes are critically important to the fitness and function of hosts. Many critically endangered plants that have been extirpated from the wild are dependent on regular fungicidal applications in greenhouses that cannot be maintained for remote out-planted populations, which quickly perish. These fungicides negatively impact potentially beneficial fungal symbionts, which may reduce plant defenses to pathogens once fungicide treatments are stopped. Using the host/parasite system of *Phyllostegia kaalaensis* and *Neoerysiphe galeopsidis*, we conducted experiments to test total foliar microbiome transplants from healthy wild relatives onto fungicide-dependent endangered plants in an attempt to mitigate disease and reduce dependency on fungicides. Plants were treated with total microbiome transplants or cultured subsets of this community and monitored for disease severity. High-throughput DNA screening of fungal ITS1 rDNA was used to track the leaf-associated fungal communities and evaluate the effectiveness of transplantation methods. Individuals receiving traditionally isolated fungal treatments showed no improvement, but those receiving applications of a simple leaf slurry containing an uncultured fungal community showed significant disease reduction, to which we partially attribute an increase in the mycoparasitic *Pseudozyma aphidis*. These results were replicated in two independent experimental rounds. Treated plants have since been moved to a native habitat and, as of this writing, remain disease-free. Our results demonstrate the effectiveness of a simple low-tech method for transferring beneficial microbes from healthy wild plants to greenhouse-raised plants with reduced symbiotic microbiota. This technique was effective at reducing disease, and in conferring increased survival to an out-planted population of critically endangered plants. It was not effective in a closely related plant. Plant conservation efforts should strive to include foliar microbes as part of comprehensive management plans.

# INTRODUCTION

So far, foliar fungal endophytes have been found in every natural plant examined (*Petrini, 1986*; *Rodriguez et al., 2009*). These fungi likely perform various functions within hosts,

Corresponding author
Anthony S. Amend,
amend@hawaii.edu

but are often defined negatively, as leaf-associated fungi that do not show pathogenicity (*Hardoim et al., 2015*). This definition is contextually dependent on a wide range of factors that influence how leaf-inhabiting fungi interact with their plant hosts, including fungal genotype (*Rudgers, Fischer & Clay, 2010*), and climatic change such as increased drought frequency (*Desprez-Loustau et al., 2006*).

The fungi that inhabit the phyllosphere are likely as important to plant health as are belowground fungi  (*Vicari, Hatcher & Ayres, 2002*; *Herre et al., 2007*; *Porras-Alfaro & Bayman, 2011*). Evidence suggests that naturally occurring fungal foliar endophytes partially determine disease severity in agricultural systems (*Xia et al., 2015*; *Ridout & Newcombe, 2016*), tropical trees (*Arnold et al., 2003*), and *Populus* models (*Busby, Peay & Newcombe, 2016*). Mechanisms for this function include antagonism or protagonism toward pathogenic species, competition for resources, and/or by altering plant host defenses. Therefore, endophytes may be most usefully thought of as modifiers of plant disease (*Busby, Ridout & Newcombe, 2015*), and/or insect herbivory (*Breen, 1994*; *Hartley & Gange, 2009*) rather than as simply transitively ''non-pathogenic''.

This perspective has led to many biocontrol efforts that (with varied success) seek to reduce disease severity by using beneficial foliar endophytes, particularly in commercially important plants (*Viterbo et al., 2002*; *Kiss, 2003*; *Miller et al., 2004*; *Bressan & Borges, 2004*; *Gafni et al., 2015*; *Borges, Saraiva & Maffia, 2015*). However, to date, there seems to be less effort to apply this knowledge to plant conservation efforts. The work that has addressed any microbial components of plant conservation has focused mostly on belowground plant-microbe relationships, especially on arbuscular mycorrhizal symbioses (e.g., *Requena et al., 2001*; *Gemma, Koske & Habte, 2002*; *Zubek et al., 2008*; *Harris, 2009*; *Ferrazzano & Williamson, 2013*; *Rigg et al., 2017*). These studies, and others, have shown that soil microbes can play a large role in plant success in a given habitat, but relatively less attention has been granted to the aboveground microbes that interact with plants.

Fungi provide important services for plant and animal conservation targets (e.g., nutrient and water liberation and uptake) (*Heilmann-Clausen et al., 2015*), but these services remain unexplored with regard to foliar fungi. Here, we examine how manipulating foliar endophytes modifies plant disease on critically endangered plants known to suffer from disease mortality, demonstrating the potential for foliar endophytes to be used in conservation.

*Phyllostegia kaalaensis* (Lamiaceae) is a plant endemic to the Waianae Mountain range on the island of O'ahu in Hawai'i. The genus *Phyllostegia,* found only in the Hawai'ian Islands, represents a radiation from presumably one introduction of an allopolyploid ancestor and are phylogenetically nested within the North American genus *Stachys* (*Baldwin & Wagner, 2010*). This ancestor has since radiated into 32 recognized species of *Phyllostegia*. Of these, 14 are listed as critically endangered (*IUCN, 2017*) and most of the others are presumed to be extinct. Currently extirpated in the wild, *P. kaalaensis* only exists as populations in two greenhouse facilities, one managed by the state of Hawai'i and one by the US Army. Although clonal propagation is readily achieved out-planting efforts have yielded no long-term success, defined by survival of at least one year and active recruitment of new plants (*Weisenberger & Keir, 2012*). In the greenhouse environment, *P. kaalaensis* is highly

susceptible to infection by the powdery mildew *Neoerysiphe galeopsidis*, which can lead to total mortality within 30 days if untreated (M Keir and G Zahn, pers. comm., 2016). This leaves greenhouse-raised plants dependent on regular applications of topical fungicide (Mancozeb, DuPont, Wilmington, DE, USA).

Dependence on fungicides is problematic for long-term restoration goals. First, continuous application is impractical for out-planted populations in remote sites. Additionally, fungicide applications can have undesirable effects on beneficial fungal endophytes (*Karlsson et al., 2014*). Thus, it is likely that the fungicide used on greenhouse-raised *P. kaalaensis* individuals are inhibiting pathogen antagonists as well as the pathogen. This might lead to plants being reintroduced to their native range with reduced colonization of potentially beneficial foliar fungi, possibly making them more susceptible to environmental pathogens or otherwise maladapted to natural environments. We hypothesized that re-establishing endophyte communities within foliar tissues would increase disease resistance and improve out-planting success.

We conducted experimental inoculations of fungi obtained from related healthy wild plants in the previous home range of *P. kaalaensis* and show that pathogen resistance can be conferred by establishing beneficial communities of endophytes in aboveground plant tissues in order to improve endangered plant survival in the wild.

## METHODS

### Experimental design and overview

The experiment tested the disease modification properties of fungal endophyte isolates and uncultured fungi from a slurry of surface-sterilized leaves obtained from wild healthy relative, *Phyllostegia hirsuta*. *P hirsuta* is another endangered mint, whose range overlaps *P. kaalaensis*, and it was chosen as a microbial donor since outplanting efforts have yielded recent success in re-establishing stable wild populations (new plant recruitment for at least one year, M Keir, pers. comm., 2016). We chose two endangered plant species, *P. kaalaensis* and *P. mollis*, as microbial recipients due to their critically endangered status and the fact that extant populations require weekly fungicide applications. The logistics of working with critically endangered plants limited the scope of the experiment. Only ∼18 individuals per species were available at a time, so we selected three treatments: a slurry of leaves from wild *Phyllostegia hirsuta* containing uncultivated fungi, a slurry of spores from eleven cultured endophyte isolates representing a readily-cultivable subset of the leaf slurry fungi, and a sterile water control.

We exposed all plants to the *N. galeopsidis* pathogen, and disease severity was observed until plant mortality. Throughout the experiment, DNA was extracted from surface-sterilized leaves to track endophytic fungal community composition. We repeated the entire experiment a second time with a new set of 18 plants in order to confirm the initial findings and to assess reproducibility. At the conclusion of both experimental rounds, we performed a final control round consisting of two treatments, a leaf slurry and a leaf slurry filtered through 0.2 um to remove fungi and bacteria, to confirm that observed effects were attributable to biota and not to phytochemicals present in the leaf slurry. In the subsections

below, we present methods that first outline plant and inoculum preparation, describe the experimental trials, and explain the workflow for wet lab work and bioinformatic analyses.

## Plant acquisition

We acquired *P. kaalaensis* and *P. mollis* individuals from the Oahu Army Natural Resources Program (OANRP) under authorization of the USFWS on the US Army's permit (TE-043638-10). Experimental plants were grown from cuttings of greenhouse individuals from 4 clonal lines and were randomly assigned to experimental groups. Plants arrived in 4-inch pots of soil-less medium (Sunshine #4; SunGro Horticulture, Agawam, MA, USA) and remained in these pots for the duration of the experiment. Though greenhouse populations are dependent on regular chemical treatments, these individuals had not been treated with fungicide or insecticide since cuttings were taken (∼8 weeks). Plants were watered from below with sterile D.I. water every other day for the duration of the trials, and humidity was passively controlled by keeping a shallow pan of sterile water open on the floor of the growth chambers.

## Inoculum and pathogen acquisition and preparation

Fungal isolates were obtained by placing small cuttings of surface-sterilized *P. hirsuta* leaves, collected from the wild, on MEA medium amended with Streptomycin and Kanamycin (Supplemental Information). After three weeks of growth, we identified 11 morphologically dissimilar sporulating isolates by Sanger sequencing of the ITS1-28S region of ribosomal-encoding DNA amplified with ITS1F (5′-CTTGGTCATTTAGAGGAAGTAA-3′) (*Gardes & Bruns, 1993*) and TW-13 (5′-GGTCCGTGTTTCAAGACG-3′) (*White et al., 1990*). Molecular identification supported the separation of the morphologically-distinct isolates. These isolate cultures were flooded with sterile water, gently shaken to release spores, and spores were pooled in equal concentrations ($2.3 \times 10^6$ cells/mL) to compose the "isolate slurry".

The leaf slurries were obtained by blending surface-sterilized *P. hirsuta* leaves in sterile water for 1 min in a Waring Laboratory Blender and then filtering through a 100 μm membrane to remove large particles. The resulting "leaf slurry" contained the natural endophytic community of *P. hirsuta* and was used without further processing.

## Incubation and pathogen challenge

Plants were kept in Percival growth chambers at 21 degrees C under 12 h of light per day (550 μmoles PAR m$^2$ s$^{-1}$) and watered twice weekly. We used a foliar spray method similar to *Posada et al. (2007)* to inoculate leaves. Briefly, inoculation was performed with a hand sprayer, applying approximately 5 ml of inoculum per plant, per application period, and plants were covered by plastic bags for 24 h immediately after to increase humidity. To improve the efficacy of any potential biocontrol agents (*Filonow et al., 1996*), plants were inoculated weekly for three weeks prior to pathogen exposure. After three weeks, the pathogen was introduced by placing an infected *P. kaalaensis* leaf from the OANRP greenhouse in the air intake of the growth chambers. Weekly, all the leaves of each plant were visually inspected for signs of infection and the total proportion of infected leaf area was recorded as a measure of disease severity.

## DNA methods

We extracted DNA from the inoculum sources and from surface sterilized leaf punches when the plants arrived, in the middle (immediately after the first visible signs of powdery mildew infection), and at the end of incubations. Two leaf punches from each plant were made with a 1 cm diameter sterile hole punch, avoiding visibly infected areas, and were surface-sterilized by shaking in 1% bleach for 1 min, 70% ethanol for 2 min, and two rinses in sterile water for 2 min each. Inoculum slurries were centrifuged for 10 min at 10,000 RCF and resultant pellets were retained for DNA extraction. DNA was extracted from surface-sterilized leaf punches and inoculum pellets with MoBio Powersoil kits (QIAGEN, Venlo, The Netherlands).

Because of rapid leaf loss and/or pathogen coverage on individuals once infected, it was not possible to always obtain two leaf disks from each plant. Therefore, for each sampling period, we pooled leaf disks within each group and randomly selected two plugs for each of three extractions.

Fungal DNA was amplified with ITS1F and ITS2 (*White et al., 1990*), modified with the addition of Illumina adaptors (*Caporaso et al., 2011*) using the following protocol: 98 2 min; 22 cycles of: 98 15 s, 52 30 s, 72 30 s; 72 2 min). After 22 cycles, the PCR product was diluted 1:12 and 1 µL of this was used as a template for 8 more rounds of PCR with a 60 deg annealing temperature in which bi-directional barcodes bound to reverse complimented Illumina adaptors acted as primers. Resulting barcoded libraries were cleaned, normalized, and sequenced with the Illumina MiSeq platform (V3 chemistry, $2 \times 300$ bp).

## Bioinformatics/Statistics

The general bioinformatics strategy consisted of bi-directional read pairing, quality filtration, and chimera removal, followed by extraction of the ITS1 region and open-reference OTU picking. Illumina reads were demultiplexed by unique barcode pairs and forward and reverse reads were merged with Pear (*Zhang et al., 2014*). Reads that were successfully assembled were then quality screened with the fastx_toolkit (http://hannonlab.cshl.edu/fastx_toolkit/index.html) to remove reads shorter than 200 bp or longer than 500 bp and those that contained any bases with a quality score lower than 25.

Quality-screened reads were then checked for chimeras both de novo and against the UNITE-based chimera database (*Nilsson et al., 2015*; downloaded 31.01.2016) to remove any putative chimeric sequences with VSearch 1.9.1 (*Rognes et al., 2016*). Non-chimeric sequences (those passing both screening steps) were subsequently run through ITSx (*Bengtsson-Palme et al., 2013*) to extract fungal ITS1 sequences (i.e., only the ITS1 region of sequences determined to be fungal in origin).

OTUs were clustered at 97% similarity from screened ITS1 sequences with the uclust algorithm (*Edgar, 2010*) wrapped within the open-reference OTU picking workflow of QIIME version 1.9.1 (*Caporaso et al., 2010*) and taxonomy was assigned against the dynamic UNITE fungal database (*Kõljalg et al., 2013*) version 1.31.2016. The resultant OTU table was then filtered in R (version 3.3.3) to remove singletons and OTUs that occurred in a given sample at less than 0.1% of the abundance of the maximum read abundance to control for index bleed-over. Finally, reads present in extraction and PCR negatives were

subtracted from samples and the OTU table was subsampled to a depth of 8,000 reads per sample with the vegan package in R (*Okansen et al., 2016*) to determine normalized relative abundance. Bray–Curtis community dissimilarity measures were performed on rarefied data with the vegdist function of the vegan package in R.

We initially identified potentially beneficial OTUs (i.e., those associated with reduced disease severity) with the indicspecies R package (*Cáceres & Legendre, 2009*) on samples grouped by quartile values into bins of disease coverage, measured as percent of leaf surface area infected. OTUs that were significantly correlated with low-disease samples were then tested as predictors of *N. galeopsidis* relative abundance and disease severity in a generalized linear model with a binomial family and logistic link function.

# RESULTS

## Disease progression and treatment effectiveness

The fungal isolate slurry treatment did not reduce disease severity in either plant species during either experimental round, whereas the wild leaf slurry reduced disease severity in *P. kaalaensis* in both trials. (Binomial GLM; Round 1: $P = 0.0029$, Pseudo-R2 $= 0.808$; Round 2: $P = 0.0015$, Pseudo-R2 $= 0.745$). The two experimental rounds showed congruent results, though on different time scales. Plants in the first round rapidly succumbed to *N. galeopsidis* infection after about 30 days, but during the second round, disease took longer to manifest with infections showing up at ∼30 days, and plant mortality by ∼90 days. *P. mollis* individuals did not respond to either treatment (Fig. 1) and are excluded from further analyses. The additional control round (performed only with *P. kaalaensis*) demonstrated that removing biota from the wild leaf slurry with a 0.2 µm filter eliminated the beneficial effects, with the unfiltered slurry showing significantly less disease severity than the filtered slurry (Binomial GLM; $P = 0.0034$).

## Bioinformatics

The sequencing run returned 2,273,484 raw forward and reverse reads for analyses. Of these, 2,136,144 were successfully merged. After quality filtering, ITS extraction, and chimera removal, 1,629,699 reads remained, yielding 199 OTUs after singleton removal. Eight OTUs accounted for ∼94% of all reads, and a single OTU (*N. galeopsidis*) accounted for ∼76% of all reads.

## Fungal communities in slurries and leaves

The vast majority of sequences from the wild leaf slurries were identified as the pathogen, *N. galeopsidis*. This was surprising, given that the *P. hirsuta* individuals donating to this slurry showed no signs of powdery mildew infection, and considering that the wild leaf slurry was the treatment shown to reduce *N. galeopsidis* disease severity. Twenty-one other OTUs were detected in the leaf slurry inoculum over both rounds, but none of these, other than *Neopestalotiopsis saprophytica*, comprised greater than 5% relative abundance (See Fig. 2). Sequence libraries of fungal isolate slurry samples contained 8 OTUs (representing 8 of the 11 isolates added to the slurry) and were similarly dominated by a single taxon, *Alternaria alternata*. Although three taxa were not recovered by sequencing, all 11 fungal taxa were successfully re-isolated from the slurry on MEA media.

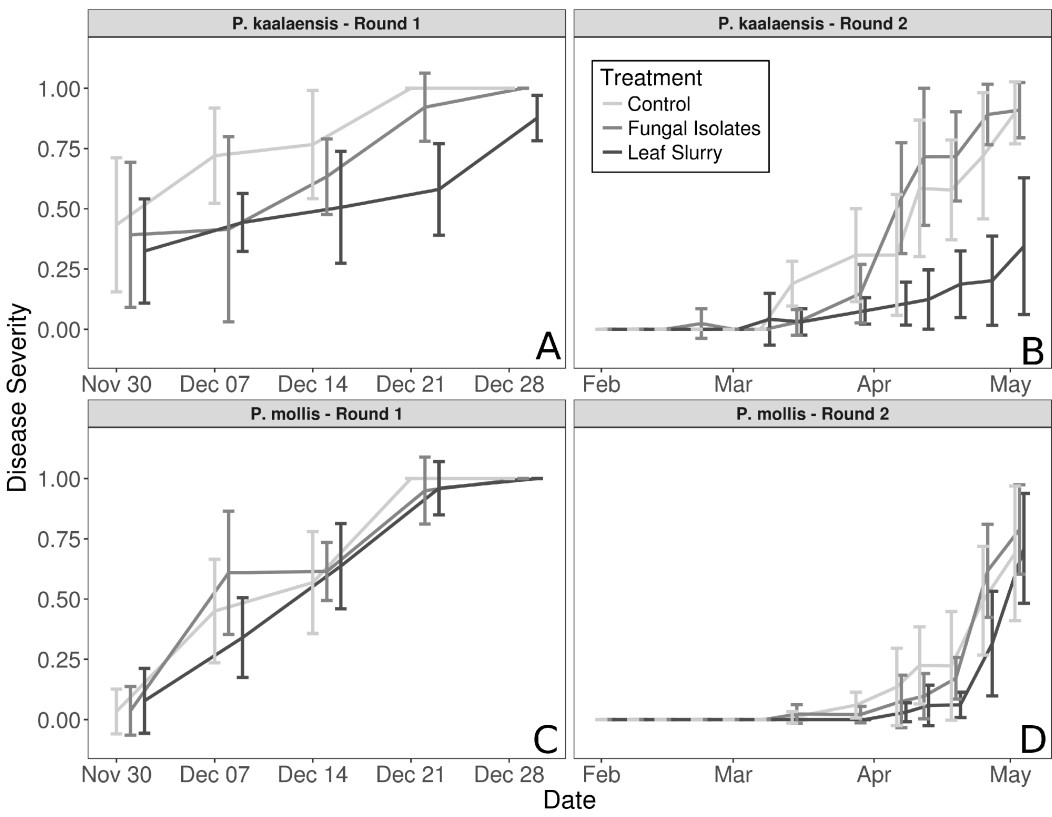

**Figure 1** **Plant disease progression for both experimental rounds.** Disease severity over time for each plant species during each experimental trial, measured as percentage of leaf area visibly infected by powdery mildew. (A and B)—*P. kaalaensis*; (C and D)—*P. mollis*. The first trial (A and C) lasted 30 days and the second trial (B and D) lasted 90 days. *P. kaalaensis* plants receiving the whole leaf slurry had delayed infections and reduced overall infection severity (black lines). Error bars represent 95% CI around the mean.

*N. galeopsidis* OTU relative abundance correlated strongly with increased disease severity in plants (Binomial GLM; $P = 0.0020$, Pseduo-R2 $= 0.75$). Both disease severity and *N. galeopsidis* relative abundance were negatively correlated with the relative abundance of a single taxon, the mycoparasitic basidiomycete yeast *Pseudozyma aphidis* (Binomial GLM; Disease Severity: $P = 0.0112$; ; *N. galeopsidis* rel. abundance: $P = 0.0071$). *P. aphidis* was found in low relative abundance in plant leaves from all treatment groups prior to experimental inoculations, but just after the first pathogen infections were visible it was significantly more abundant in plants receiving the wild leaf slurry. Individuals with greater relative abundance of *P. aphidis* showed sharply reduced infection severity (Fig. 3).

Eleven OTUs (other than *N. galeopsidis*) transferred from the leaf slurry onto plant leaves were still detected halfway through the growth periods, while only six were detected at the end of the study. Pathogen infection load similarity was a strong driver of community similarity (ANOVA: $P < 0.00005$, R2 $= 0.481$). Plants with very high and very low infection severities hosted fungal communities that were more similar than plants with intermediate infection severities. Though this was temporally confounded (infection severity and time

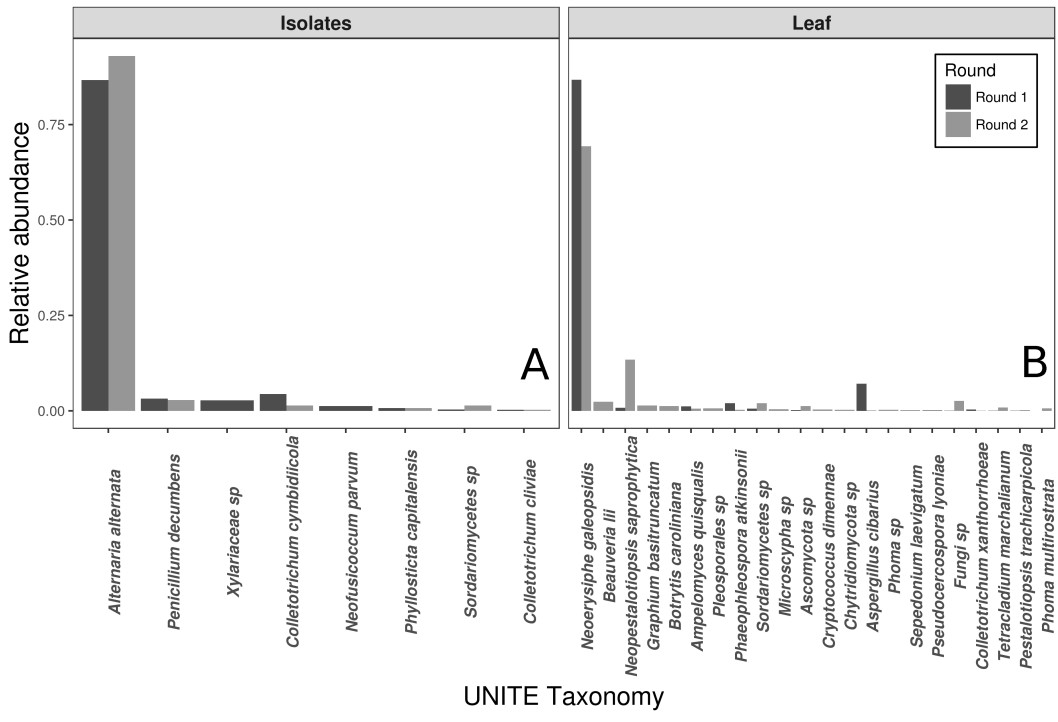

**Figure 2** **Taxonomic compositions of the two experimental donor inoculae.** Species compositions (relative abundance) for each inoculum treatment, during both experimental trials. Both the fungal isolates (A) and the whole leaf slurries (B) were dominated by a single taxon. Taxonomy reflects assignments to the UNITE fungal ITS database.

are not independent) the trend toward community convergence was driven largely by *N. galeopsidis* proliferation and infection (Figs. S1 and S2).

## Outplanting

Six healthy *P. kaalaensis* individuals from the leaf slurry treatment showing no sign of pathogens were out-planted in April 2016 in a native habitat for monitoring. As of August 2017 they have remained disease-free, and are now the only extant population of *P. kaalaensis* in the wild. The out-planting site is less than 1 km from the location of the *P. hirsuta* from which we obtained the leaf slurry inoculum, so it is presumed that there are ample *N. galeopsidis* propagules locally. This reinforces the conclusion that the microbiome transplantations are serving to protect out-planted individuals from the pathogen.

## DISCUSSION

This study demonstrates that foliar endophytes modify plant disease, and can be used in endangered plant conservation, much as they have been for agriculturally important plants. The low-tech method of spraying plants with a slurry of leaves from healthy wild relatives (containing many uncultured/unculturable fungal taxa) outperformed inoculations of fungal isolates, suggesting that biodiversity was important for the functional relevance of the inoculated microbes.

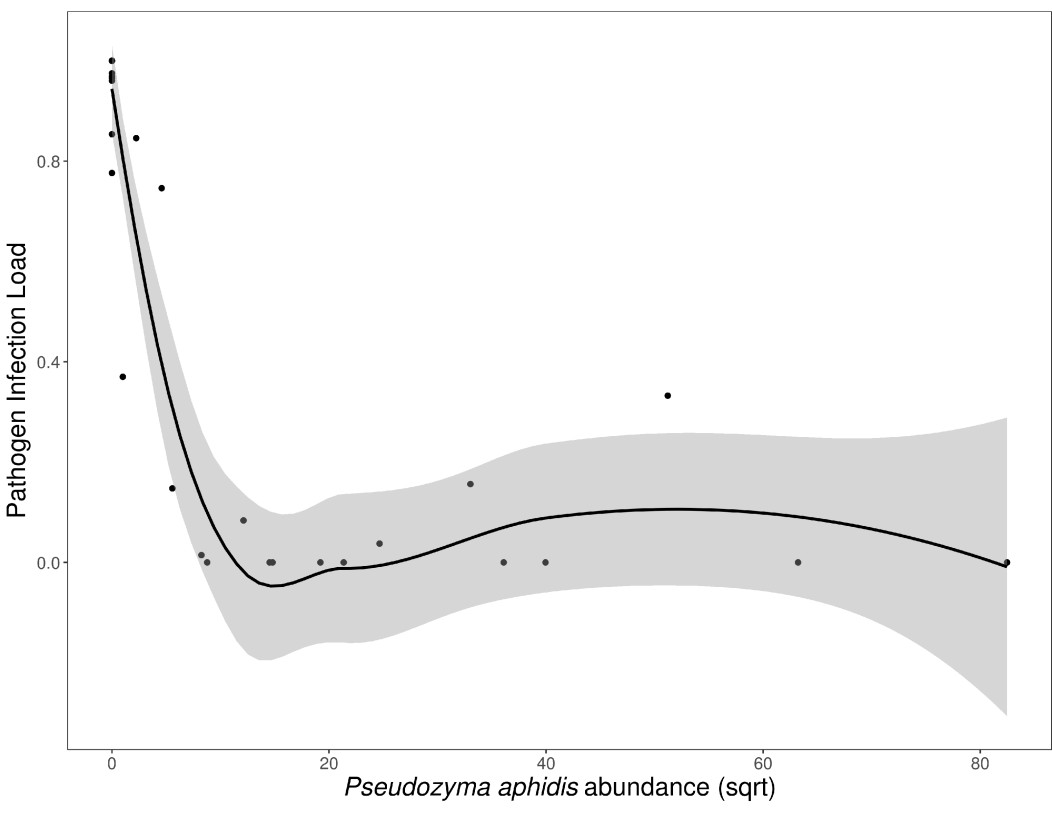

**Figure 3** **Disease severity as a function of *P. aphidis* abundance.** *N. galeopsidis* infection severity as a function of *P. aphidis* abundance. Higher *P. aphidis* abundance was negatively correlated with infection severity. This figure shows data from *P. kaalaensis* observations over both experimental rounds. Line represents loess smoothing and gray area represents 95% CI around the mean.

The leaf slurry treatment reduced disease severity in plants despite that the very pathogen we were trying to mitigate dominated the sequencing library. The donor plants showed no obvious signs of *N. galeopsidis* infection, and it is possible that the strain present in the slurry differed from the strain causing *P. kaalaensis* mortality. However, we were unable to determine this from our data since all ITS1 reads assigned to *N. galeopsidis* were nearly identical to the voucher sequence for the pathogen found on Oahu (with the exception of four singleton variants that differed slightly but were removed because each only had one read; see Supplemental Information). Further, the ITS1 reads assigned to *N. galeopsidis* from both the slurry and infected plants were identical. *N. galeopsidis* is known to cause disease in all studied species of *Phyllostegia,* in the ancestral genus *Stachys* within North America, (*Glawe, 2007*) and a closely related strain of *N. galeopsidis* has been reported on *Stachys* hosts from Eastern Asia (*Heluta et al., 2010*). It is not known how recently *N. galeopsidis* arrived in Hawai'i or whether it came with the original ancestor to modern *Phyllostegia* species.

The relative abundance of the mycoparasitic fungus, *P. aphidis*, is a plausible explanation for the decrease in disease severity since *P. aphidis* has previously been shown to be

antagonistic against powdery mildews (*Buxdorf, Rahat & Levy, 2013*; *Gafni et al., 2015*), and to reduce the incidence of plant disease (*Barda et al., 2015*). Its genome contains genes for chitinase, two chitinase-related genes, and other cell-wall degrading proteins (*Lorenz et al., 2014*). Additionally, it has been shown to promote plant health, possibly via heavy siderophore production which potentially limits pathogen growth by chelating available iron (*Fu et al., 2016*).

It is possible that other fungi or other microbes contributed to our observed pattern, including epiphytic species. In fact, *P. aphidis* was not detected in the slurry extracts from either experimental round, meaning that it was either not present or that it was present at undetectably low relative abundance given the numerical dominance of *N. galeopsidis* reads. In this case, it seems likely that a diverse assemblage of fungi (and/or bacteria) was responsible for the relative increase in *P. aphidis* relative abundance in plants sprayed with the leaf slurry. At the end of the study, light microscopy of necrotic lesions taken from plants treated with the leaf slurry appeared to reveal *P. aphidis* attacking *N. galeopsidis* spores, though this is not conclusive (See Supplemental Information). The ability to determine the success of comprehensive fungal microbiome transplantations was limited by the dominance of the pathogen in final amplicon sequences. Taxa with low relative abundances were less likely to be detected as *N. galeopsidis* reads proliferated at the end of the growth periods, but the 11 taxa that were detected halfway through the trial and 6 that were detected at the end indicate that the simple indiscriminate transplantation of leaf microbiota was successful in establishing a diverse assemblage of uncultured endophytes.

Primer biases or preferential Illumina clustering for shorter sequences were potentially responsible for the dominance of *A. alternata* in isolate inoculum reads, despite spore-count normalization. *Adams et al. (2013)* demonstrated that abundant fungal ITS reads have the potential to swamp out known community members. We did not observe the same ameliorative effect of *P. aphidis* in the other endangered plant species, *P. mollis*, which implies that plant-microbe interactions were important for our outcome. *Barda et al. (2015)* showed that *P. aphidis* was capable of inducing pathogenesis-related genes and triggering an induced pathogen resistance response in tomato plants and it is possible that induced host plant defenses instead of, or in addition to, direct antagonism played a part in the positive outcome for *P. kaalaensis*. Further, it is possible that other microbial taxa, such as bacteria, were instrumental in producing the ameliorative effect of the leaf slurry.

This study reinforces the idea plants are not just plants; they are a complex assemblage of organisms (*Porras-Alfaro & Bayman, 2011*), and should be considered as such when planning conservation approaches. Since they are integral components of plant health, foliar fungi should be a key aspect of management plans for endangered plants, particularly those suffering from pathogen-induced mortality. This simple approach of wholesale transplantation of a microbiome conferred disease resistance to a plant on the brink of extinction, and may be usefully applied to other plants.

## ACKNOWLEDGEMENTS

We would like to gratefully acknowledge the US Army for logistical support: K Kawelo, M Kier, L Weisenberger and V Costello from US Army Garrison - Hawai'i's O'ahu Army

Natural Resources Program (OANRP) for invaluable field and greenhouse assistance, for their expertise in rare plant conservation, and for providing the plants used in this study, along with B Sedlmayer for assisting with disease monitoring.

### Funding

This project was funded through the US Army cooperative agreement W9126G-11-2-0066 with Pacific Cooperative Studies Unit and the NSF DEB-1255972. The funders had no role in study design, data collection and analysis, decision to publish, or preparation of the manuscript.

### Grant Disclosures

The following grant information was disclosed by the authors:
US Army cooperative agreement: W9126G-11-2-0066.
NSF: DEB-1255972.

### Competing Interests

Anthony Amend is an Academic Editor for PeerJ.

### Author Contributions

- Geoffrey Zahn conceived and designed the experiments, performed the experiments, analyzed the data, wrote the paper, prepared figures and/or tables, reviewed drafts of the paper.
- Anthony S. Amend conceived and designed the experiments, analyzed the data, contributed reagents/materials/analysis tools, wrote the paper, reviewed drafts of the paper.

### Field Study Permissions

The following information was supplied relating to field study approvals (i.e., approving body and any reference numbers):

Work with endangered plants was made possible under the US Fish and Wildlife Service Transfer Agreement TE-043638-10.

### DNA Deposition

The following information was supplied regarding the deposition of DNA sequences:

Raw Illumina sequences of ITS amplicons used in this study have been deposited in the Sequence Read Archive; BioProject Accession: PRJNA342669. Sanger sequences of fungal isolates have been deposited in Genbank: KX988291–KX988301.

### Data Availability

The raw data and code have been provided as Supplemental Files.

## Supplemental Information

Supplemental information for this article can be found online at http://dx.doi.org/10.7717/peerj.4020#supplemental-information.

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
