# Peer review of "Foliar microbiome transplants confer disease resistance in a critically-endangered plant"

_PeerJ, doi:10.7717/peerj.4020_

## Round 0.1 · original submission · Minor Revisions

· Academic Editor

Minor Revisions

These are interesting findings and I agree with the reviewers that the manuscript would be deepened and improved by providing more more of the biology/natural history of the systems studied. Two reviewers would like some additional details and discussion on the finding of the powdery mildew in the slurry. Please consider all reviewer comments for the purposes of clarity.

·

Basic reporting

This is a good manuscript.

Experimental design

No problem. See General Comments.

Validity of the findings

No problem. See General Comments.

Additional comments

This is a well written manuscript on interesting results. I don’t see any issues with design and I think that the findings are valid. So, I’ll proceed with more general comments to hopefully help the authors find ways to improve the manuscript. I think that the context could be deepened somewhat here and there. A brief background is needed on species of Phyllostegia endemic to Hawaii and other Pacific islands. Is there any knowledge of what continental plant gave rise in the islands to endemic Phyllostegia, and of when that migration event might have occurred? Is that ancestor a host to this same powdery mildew, at least at the taxon level? A little more context and background on the powdery mildew might be helpful too. It is considered to be currently cosmopolitan but is there any reason to think that the mildew was recently introduced? Or does it seem more likely that the mildew has always been there in the islands as diversification in Phyllostegia progressed? I am not aware of any endophyte study reporting that the “ vast majority of sequences from … were identified as the pathogen”. Might the authors draw a little more attention to this finding if it truly is unique? The authors report relatively little endophyte diversity, and might emphasize that more. Another question that should be raised has to do with the presence in the leaf slurries of Ampelomyces quisqualis. The latter is a known mycoparasite of powdery mildew. Was the mildew that developed on plants examined microscopically for the presence of pycnidia of Ampelomyces? Was it examined for the yeast?

·

Basic reporting

It was difficult to get hooked into the manuscript details.

There was no indication of name of the plant species until Line 65 or the causal agent of the disease in Line 70. I suggest these organisms be identified in the title and abstract to keep the reader engaged.

Why did the authors choose this pathosystem to study? What is significant about this plant other than it is endangered, many plants species have this status.

The continued use of the generic terms 'pathogen' and 'plant' is vague. The presented research focuses on specific plant species, one specific powdery mildew pathogen species, a specific and potentially important yeast biocontrol agent and a few named fungal taxa in the discussion. The research has potential implications for the conservation of highly imperiled plant species. To sustain the reader's interest, it may be best to relate the pertinent information using clear communication of the taxa involved throughout the manuscript.

In Line 63, the use of the term 'pathogen mortality' is not clear. Does the pathogen die or does the host plant die as a result of colonization by the pathogen?

In Lines 61, 128, 130 and 165 the authors use quotation marks. When one is unsure of the correct term, it may be preferred to choose words that convey the intended meaning to facilitate comprehension by the reader rather than highlight perceived indecisiveness with quotation marks. If these terms are in fact quotations from another source, then a citation will clarify the use of quotation marks.

In Lines 97-98, 116, 146, 147-148, 158, 159, 161, 172, 199, 227, 251-252 and 265 the authors use parentheses. This is unconventional and discredits the strength of the statements.

Experimental design

Lines 139-140, the authors discuss 'pathogen exposure and pathogen introduction'. Is this in reference to N. galeopsidis?

Continuing on in Lines 140-142 the authors mention the air intake system in the greenhouse. Are you claiming to inoculate your experiment through random chance of air dispersal through the greenhouse ventilation system? If so, this unusual inoculation method will be best supported with citations of its use in other peer-reviewed publications.

Were negative controls used in this experimental design? If so, it will be important to report the data collected for comparison to the inoculated plants.

Validity of the findings

The research presented claims to be preliminarily successful in the use of a microbiome inoculation on the leaf tissue of Phyllostegia hirsuta to protect the plant from disease after out-planting. This is an important finding, if there is evidence the pathogen is present at the out-planting site to support the claim the slurry inoculation technique shows true efficacy.

I strongly urge the authors to become familiar with the biology of the powdery mildew organism and its relationship with a host plant. Powdery mildew infections are almost always host specific. Therefore, P. hirsuta is in all likelihood able to defend against successful colonization by Neoerysiphe galeopsidis even though its close relative, P. kaalaensis is deemed to be less resistant to infection by this pathogen. Moreover, powdery mildew conidia burst in free water e.g., sterile water slurry. The OTU counts of N. galeopsidis the authors obtained in the sequencing effort using the P. hirsuta leaf tissue may represent dead cells of N. galeopsidis rather than viable cells. As the manuscript reads now, the analysis of the experimental results does not give a convincing argument for the underlying biology of the N. galeopsidis and its interaction with P. hirsuta.

Additional comments

One interesting finding is the potential biocontrol interaction between Pseudozyma aphidis and N. galeopsidis. This yeast has been shown to be an effective biocontrol agent against Podosphaera xanthii on cucumber. Gafni et al. (Frontiers in Plant Science, 2015, doi:10.3389/fpls.2015.00132).

Reviewer 3 ·

Basic reporting

The manuscript by Zahn and Amend describes the efficacy of using foliar microbiome extracts on the endangered Phyllostegia spp. as biocontrol agents against the foliar pathogen N. galeopsidis in Hawai’i. The manuscript examines two endangered species. P. kaalaensis and P. mollis, that received inoculations of either fungal endophyte spores or filtered leaf extracts prior to inoculation with the pathogen. The foliar fungal community is monitored at three intervals with NGS amplicon sequencing of the ITS rDNA locus.

The manuscript is written clearly and the authors provide sufficient background and context to understand the rationale for the study, the methods, and the results. The figures are professional and easy convey the results. The raw data are shared in an appropriate manner.

Experimental design

The research question is well defined and meaningful. In general the methods are clearly described in sufficient detail for the reader to replicate the study. I have noted below one area where additional detail on methodology is necessary to replicate the results.

Validity of the findings

The experimental methods and data analysis are experimentally robust and the author's conclusions are well stated and supported by the results of the study.

Additional comments

I have some general comments and suggestions.

Line 22. ‘Throughput’ is misspelled.
Line 38. To help clarify the meaning of this sentence change ‘and are often defined negatively’’ to ‘but are often defined negatively’.
Line 125. Please provide citations for primers use in the study (ITS1F and TW-13).
Line 129. What method was used for blending? Please describe in greater detail.

---

## Round 0.2 · Minor Revisions

· Academic Editor

Minor Revisions

I apologize for sending the manuscript back again, but I noticed that a few of the references cited in the text are missing from the references section. Please check the references, make the additions, and I will accept your revision in a timely manner. I have highlighted the ones I noticed in the attached document as well as several typos. I also recommend you use "disease mortality" instead of "pathogenic mortality" on line 62.

---

## Round 0.3 · accepted · Accept

· Academic Editor

Accept

Thank you for your edits. Congratulations on an interesting study!